

# Nano-sized zeolites as modulators of thiacloprid toxicity on *Chironomus riparius*

Carla S. Lorenz[1], Anna-Jorina Wicht[2], Leyla Guluzada[3], Barbara Crone[4], Uwe Karst[4], Hwa Jun Lee[5], Rita Triebskorn[1,6], Stefan B. Haderlein[3], Carolin Huhn[2] and Heinz-R. Köhler[1]

[1] Institute of Evolution and Ecology, Animal Physiological Ecology, Eberhard-Karls-Universität Tübingen, Tübingen, Germany
[2] Institute of Physical and Theoretical Chemistry, Eberhard-Karls-Universität Tübingen, Tübingen, Germany
[3] Center for Applied Geosciences, Environmental Mineralogy and Chemistry, Eberhard-Karls-Universität Tübingen, Tübingen, Germany
[4] Institute of Inorganic and Analytical Chemistry, Westfälische Wilhelms-Universität Münster, Münster, Germany
[5] Center for Ordered Nanoporous Materials Synthesis, Division of Environmental Science and Engineering, Pohang University of Science and Technology, Pohang, South Korea
[6] Steinbeis Transfer-Center for Ecotoxicology and Ecophysiology, Rottenburg, Germany

Corresponding author
Carla S. Lorenz, lorenz.carla@web.de

## ABSTRACT

This study investigated whether zeolites of different size (Y30 (nano-sized) and H-Beta(OH)-III (forming large aggregates/agglomerates composed of 50 nm small primary particles)) exerted acute toxicity on larvae of the non-biting midge, *Chironomus riparius*, and whether such zeolites are able to modulate the toxicity of a common insecticide, thiacloprid, by means of adsorption of a dissolved toxicant. We conducted acute toxicity tests with fourth instar larvae of *C. riparius*. In these tests, larvae were exposed to zeolites or thiacloprid solely, or to mixtures of both compounds. The mixtures comprised 1.0 µg/L thiacloprid in addition to low (5.2 mg/L), medium (18.2 mg/L), and high (391.7 mg/L) zeolite concentrations, resulting in different adsorption rates of thiacloprid. As biological endpoints, changes in mortality rates and in behavior were monitored every 24 h over a total investigation period of 96 h. Furthermore, we conducted chemical analyses of thiacloprid in the medium and the larvae and located the zeolite particles within the larvae by LA-ICP-MS imaging techniques. Our results demonstrate that both types of zeolites did not exert acute toxicity when applied as single-substances, but led to reduced acute toxicity of thiacloprid when applied together with thiacloprid. These results are in line with the sorption properties of zeolites indicating reduced bioavailability of thiacloprid, although our data indicate that thiacloprid can desorb from zeolites to some extent. While freely dissolved (i.e., non-sorbed) fraction of thiacloprid was a good parameter to roughly estimate toxic effects, it did not correlate with measured internal thiacloprid concentrations. Moreover, it was shown that both zeolite types were ingested by the larvae, but no indication for cellular uptake of them was found.

## INTRODUCTION

The toxicity of particles within the nanoscale range has been intensively investigated during the last years, due to their unique properties and industrial benefits. While early studies addressed "colloids" up to 1 µm (*Everett, 1972*) the focus has shifted towards "nanoparticles" with particles sizes of 1–100 nm (e.g., *Delay & Frimmel, 2012*; *Hartland et al., 2013*; *Lead & Wilkinson, 2006*; *Nowack & Bucheli, 2007*; *Scown, Van Aerle & Tyler, 2010*). The origin of colloids/nanoparticles can be natural or synthetic (e.g., *Delay & Frimmel, 2012*; *Lead & Wilkinson, 2006*; *Navarro et al., 2008*; *Nowack & Bucheli, 2007*; *Oberdörster, Oberdörster & Oberdörster, 2005*). Particularly synthetic particles have nowadays widespread applications, for example in electronics, personal care products, or pharmaceuticals (e.g., *Handy & Shaw, 2007*; *Nowack & Bucheli, 2007*; *Oberdörster, 2010*; *Oberdörster, Oberdörster & Oberdörster, 2005*; *Savolainen et al., 2010*). Due to the numerous application fields, such particles enter the environment intentionally and unintentionally (e.g., *Delay & Frimmel, 2012*; *Maynard, 2006*; *Moore, 2006*; *Navarro et al., 2008*; *Nowack & Bucheli, 2007*; *Wagner et al., 2014*) and thus potentially may lead to detrimental effects on organisms. For example, nanoparticles were shown to cause stress in biota by inducing reactive oxygen species (*Navarro et al., 2008*; *Savolainen et al., 2010*), by affecting cell membranes (e.g., *Bacchetta et al., 2012*; *Moore, 2006*; *Schultz et al., 2015*) or by increasing the uptake of chemicals by acting as a vehicle (*Moore, 2006*; *Nowack & Bucheli, 2007*; *Scown, Van Aerle & Tyler, 2010*). Therefore, nanoparticles became a "hot topic" in ecotoxicology. Generally, risk assessment of colloids/nanoparticles is very complex (e.g., *Delay & Frimmel, 2012*; *Handy & Shaw, 2007*; *Maynard, 2006*; *Navarro et al., 2008*; *Oberdörster, 2010*; *Oberdörster, Oberdörster & Oberdörster, 2005*; *Savolainen et al., 2010*), partly because adequate analytical methods to quantify nanoparticles were scarce. Consequently, little is known about the environmental concentrations and fate of such particles (*Delay & Frimmel, 2012*; *Maynard, 2006*; *Navarro et al., 2008*; *Oberdörster, 2010*; *Scown, Van Aerle & Tyler, 2010*). General predictions cannot be made, because since particles—even if chemically identical—may vary in size, surface characteristics or shape, and the herewith related properties (*Moore, 2006*; *Navarro et al., 2008*; *Oberdörster, 2010*; *Oberdörster, Oberdörster & Oberdörster, 2005*; *Savolainen et al., 2010*; *Scown, Van Aerle & Tyler, 2010*). Moreover, colloids/nanoparticles can interact with environmental pollutants, other particles or biota (e.g., *Delay & Frimmel, 2012*; *Lead & Wilkinson, 2006*; *Navarro et al., 2008*; *Nowack & Bucheli, 2007*; *Savolainen et al., 2010*; *Scown, Van Aerle & Tyler, 2010*).

In our study, we focused on the role of colloids/nanoparticles as sorbents for toxic chemicals. While adsorption to nanoparticles could lead to enhanced exposure and thus toxicity of chemicals, e.g., by colloid-facilitated transport into organisms, adsorption might also cause reduced bioavailability and thus decrease the toxicity of chemicals (*Baun et al., 2008*; *Hartland et al., 2013*; *Lead & Wilkinson, 2006*; *Navarro et al., 2008*; *Nowack & Bucheli, 2007*). In our study, we investigated the effects of adsorption on the toxicity of the neonicotinoid insecticide thiacloprid in the presence of nano-sized zeolites as sorbents. We tested two zeolite types with different particle sizes because the size of the adsorbent might influence the effective toxicity of the adsorbate. Zeolites are highly suitable materials

to address these questions as they are strong adsorbents and environmentally relevant due to applications in various fields, e.g., as catalysts in the chemical industry, as odor control compounds, in (waste–)water treatment to remove organic pollutants and heavy metals and as drug carriers (e.g., *Braschi et al., 2010*; *Ellis & Korth, 1993*; *James & Sampath, 1999*; *Lehman & Larsen, 2014*; *Ötker & Akmehmet-Balcioğlu, 2005*; *Wang & Peng, 2010*; *Yilmaz & Müller, 2009*). However, toxicity studies so far mainly focused on micro-sized zeolites, whereas studies on nano-zeolites are scarce (*Lehman & Larsen, 2014*). Consequently, in this study we focused on the acute toxicity of nano-sized, highly sorptive zeolites (Y30 and H-Beta(OH)-III) in the presence and absence of thiacloprid. The tests for acute toxicity were performed with fourth instar larvae of the non-biting midge *Chironomus riparius* in aqueous microcosms with exposure to either suspended zeolites or dissolved thiacloprid or both. The major questions addressed in our study were: (1) Do zeolites *per se* affect the mortality rate or the behavior of *C. riparius* larvae under acute exposure? (2) Do zeolites affect the toxicity of thiacloprid on larvae of *C. riparius* under acute exposure? If yes, do they act as vehicles and enhance the uptake and thus the acute toxicity of thiacloprid? Or is the bioavailability and, therefore, also the toxicity of thiacloprid reduced? (3) Does the particle size of the zeolite sorbents matter?

## MATERIAL AND METHODS

### Zeolites

Zeolites are microporous, crystalline aluminosilicates, abundant in numerous natural and synthetic modifications with the generalized formula $M_{x/n}[(AlO_2)_x(SiO_2)_y] \cdot z\ H_2O$ ($n$ = charge of M) (*Baerlocher, Olson & Meier, 2001*). Their framework consists of $AlO_4^-$—and $SiO_4$—tetrahedra with aluminum- and silica-atoms interconnected through oxygen-bridges (*Masters & Maschmeyer, 2011*). Zeolites show a very homogeneous pore size distribution with a high (up to 1,000 m²/g) internal pore surface area. Due to their highly microporous structure and structural anionic exchange sites, zeolites serve as cation exchangers and molecular sieves.

The structure of the zeolite Y30 (Thermo Fisher Scientific, USA) is of faujasite type. Faujasite type zeolites consist of sodalite cages and their pores are orthogonal to each-other. The surface area of our Y30 zeolites is 780 m²/g, the $SiO_2/Al_2O_3$ mole ratio is 30:1 and the pore volume is 0.56 cm³/g (based on $N_2$ physisorption). The average pore-diameter of Y-type zeolite is 0.74 nm and the size of a unit cell is ∼3 nm, depending on Si/Al ratio, concentration of counter-cations and degree of hydration (*Scherzer, 1978*). To obtain nano-sized particles we milled the purchased Y30 particles using a planetary mill (Pulverisette 5; Fritsch GmbH) for 30 min in zirconium oxide beakers using zirconium oxide milling balls of 0.1 mm diameter (Retsch GmbH). After dry sieving through a 0.063 mm grid to recover the milling balls, the milled zeolite was washed thrice in distilled water and methanol and centrifuged (Herolab HiCen 21) for 10 min at 5,000 rpm to eliminate larger particles by sedimentation. Sedimentation time and sedimentation velocity were calculated using Stoke's law (*Batchelor, 2000*).

Beta(OH)-III zeolites were synthesized at the Center for Ordered Nanoporous Materials Synthesis, POSTECH (Korea). The reagents for the Beta(OH)-III zeolites synthesis included

tetraethylammonium hydroxide (TEAOH, 35% aqueous solution, Aldrich), aluminum metal (99.5%, Alfa Aesar), and fumed silica (100%, Degussa). The aluminum metal source was first mixed with a solution of TEAOH in water. Then, fumed silica was added to this mixture. The chemical composition of the synthesis mixtures used here was $43.2TEAOH \cdot 0.8Al_2O_3 \cdot 80SiO_2 \cdot 1200H_2O$ (*Camblor, Corma & Valencia, 1998*). After stirring at room temperature for 24 h, the synthesis mixture was loaded into a Teflon-lined 45 mL autoclave and heated at 140 °C for 6 d with agitation (60 rpm) under autogenous pressure. The solid products were then recovered by centrifugation (15,000 rpm for 10 min), washed repeatedly with distilled water, and dried overnight at room temperature. The resulting particles with a size of 50 nm were calcinated in air at 500 °C for 8 h to remove residues of the organic template TEAOH and were refluxed twice in 1 mol/L ammonium nitrate aqueous solution at 80 °C for 6 h. After a secondary calcination in air at 550 °C for 2 h the particles (H-Beta(OH)-III) were obtained for further tests.

### Size measurements and sorption experiments

Particle sizes of both zeolites were measured by a laser diffraction method in aqueous suspension. Zeolites were dispersed in MilliQ water (MilliQ, academic, Filter: Millipak 40 gamma gold 0.22 µm) via ultrasonication (2 min) and analyzed in a Mastersizer 2000 (Malvern Instruments, Malvern, UK) with a Hydro 2000 S dispersing module (Malvern Instruments, Malvern, UK). We measured the particle size distributions of purchased Y30 particles, milled Y30 particles, Y30 particles after milling and settling, Beta(OH)-III particles and calcinated H-Beta(OH)-III particles. Moreover, we characterized the milled Y30 particles in the supernatant after sedimentation and calcinated H-Beta(OH)-III particles by SEM images using a Hitachi SU 8030 scanning electron microscope, operated at an accelerated voltage of 1 kV. For this purpose, samples were sonicated in ethanol before they were transferred to the surface of a silicon wafer.

To evaluate the sorption characteristics and to calculate the $K_d$ value of Y30, batch experiments using aqueous suspensions of zeolites and dissolved thiacloprid were conducted at controlled conditions in compliance with exposure conditions of the acute toxicity test. Thiacloprid (1, 2, 4, 6, 8, 10, 12, 16, 20, 24 and 30 mg/L) and Y30 (0.1 and 0.25 g/L) in 0.1 mM KCl were equilibrated on an overhead shaker. Experiments were conducted at pH 7 without adding buffers, and each sample was prepared in triplicates. After 12 h of shaking, samples were centrifuged at 6,240 g (Heraeus Megafuge 1.0 R Centrifuge, at 6,000 rpm for 10 min; Thermo Fisher, Waltham, MA, USA). The supernatant was transferred to 1.5 mL autosampler glass vials and was analyzed using HPLC-UV (isocratic mode; mobile phase: MeOH 60%/$H_2O$ 40%; retention time of thiacloprid: 5 min 36 s). To determine the sorption kinetics, samples were taken and analyzed at different times (ranging from 1 min to 48 h). The extent of adsorption was calculated by a mass balance based on the difference of the initial and equilibrated aqueous thiacloprid concentrations. The sorption isotherms and the sorption coefficient ($K_d$ value, which gives the ratio between sorbed thiacloprid concentration and the concentration of thiacloprid in water. High $K_d$ values indicate that more thiacloprid is bound to the particles, which means higher sorption) of thiacloprid at Y30 was than calculated by taking an average of triplicates.

## Test concentrations and preparation of test solutions

We chose three concentrations of Y30 zeolite (low, medium and high) to obtain 30, 60 or 97% sorption of the total thiacloprid, as calculated by the following equation:

$$r_{SW} = \frac{\frac{1}{f_W} - 1}{K_d} \tag{1}$$

Where

$r_{SW}$ —concentration of adsorbent (mass of sorbent per volume of aqueous phase)

$f_W$ —fraction of the adsorbate present in the water phase

$K_d$ —distribution coefficient.

Thus, the applied zeolite concentrations were 5.2 mg/L (low), 18.2 mg/L (medium) and 391.7 mg/L (high), respectively, and we used these concentrations ($\pm 5\%$ handling variability) also for the experiments with H-Beta(OH)-III particles. For each of the two zeolite sorbents, we tested the three particle concentrations solely, as well as in mixtures with 1.0 µg/L thiacloprid. Moreover, the effects of 1.0 µg/L thiacloprid solely were assessed and a control treatment with filtered and dechlorinated tap water (filtered by iron and active carbon filter) was run simultaneously. Consequently, eight different treatments were tested within one experiment, with $n = 15$. Test concentrations were prepared with filtered and dechlorinated tap water directly before the experiment started. A stock solution of 5.0 mg/L thiacloprid (analytical standard; Sigma Aldrich, Darmstadt, Germany) in demineralized water was prepared prior to testing and was stirred overnight in the dark at 7 °C. Y30 zeolites were available in an aqueous stock suspension of 2.32 g/L zeolites dispersed in aqua dest.. This stock suspension was stirred via ultrasonication (15 min) before it was diluted to test concentrations. H-Beta(OH)-III zeolites suspensions were prepared directly from dry zeolite powder and were dispered in the test medium by ultrasonication (15 min). After setting up the test concentrations for the experiments, all dispersions were left to rest for 1 h to allow adsorption of thiacloprid to zeolites in the mixture treatments.

## Maintenance of *Chironomus riparius*

*Chironomus riparius* (Diptera, Chironomidae) belongs to one of the most abundant insects in freshwater systems and is of high ecological relevance (*Armitage, Cranston & Pinder, 1995*; *Langer-Jaesrich, Köhler & Gerhardt, 2010*). Larvae of this species are sediment-dwelling and feed on detritus (*Armitage, Cranston & Pinder, 1995*). Moreover, this midge is a well-established model organism in ecotoxicology (e.g., *OECD, 2004a*; *OECD, 2004b*; *OECD, 2010*; *OECD, 2011*) and toxic effects of thiacloprid on this organism have been described already (*Langer-Jaesrich, Köhler & Gerhardt, 2010*; *Lorenz et al., 2017*).

The animals of our study originated from a stock culture established at our laboratory at the University of Tübingen (Germany), originating from a *C. riparius* larvae stock obtained from Goethe University Frankfurt/Main (Germany) in 2013. The animals were kept in a climate chamber at a temperature of 21.0 $\pm$ 0.5 °C and a light–dark cycle of 16:8 h. Larvae were reared in basins (30 × 55 × 12 cm), containing a 3 cm thick layer of fine quartz sand (particle size: 0.1–0.3 mm) and filtered and dechlorinated tap water with a continuous, gentle aeration. They were fed every second day with ground TetraMin® fish flakes

(Tetra, Melle, Germany), and 50% of the water volume was changed once per week. Moreover, all basins were covered by breeding cages ($55 \times 65 \times 120$ cm; mesh material with a mesh size of 0.5 mm$^2$) to allow swarming and mating of adult midges.

## Acute toxicity test

Test vessels (glass, diameter: 7.0 cm, height: 6.5 cm) were saturated overnight with the corresponding test solution, even though only low, if any, adsorption of thiacloprid on glass was expected due to the chemical properties of this compound. Earlier studies under the same experimental conditions did not reveal any adsorption of thiacloprid to glass (*Lorenz et al., 2017*). As glass (SiO$_2$) is slightly negatively charged at the pH applied ($7.8 \pm 0.2$) we also exclude the zeolites (zeta potential Y30: $-42$ mV; Beta(OH)-III. $-40$ mV) to bind the surface of the vessels. Subsequently, the test vessels the test vessels were emptied before 30 g (dry wt.) fine quartz sand (SiO$_2$ particle size: 0.1–0.3 mm; adsorption of zeolites and thiacloprid was negligible for the above-mentioned reasons) and 100 mL of the respective test solution were filled into them. Prior to filling the vessels, all dispersions and solutions were stirred by ultrasonication for 15 min before they were partitioned. So, zeolites were supposed to be equally distributed within the dispersions and, consequently, also in test vessels. To adjust the temperature of the test dispersions/solutions to those of the breeding basins, we incubated the test vessels for 2 h in the climate chamber before the temperature of the medium was measured and larvae were added. Five individuals of fourth instar larvae of *C. riparius* were added to each test vessel using a blunt glass pipette. The test vessels were covered with perforated Parafilm® (Carl Roth GmbH, Karlsruhe, Germany) to minimize evaporation and placed in a random arrangement on a desk within the climate chamber. Larvae were exposed for 96 h and no food was added to avoid potential interactions between food and test substances. Every 24 h we checked the test vessels and collected data on mortality (i.e., the absence of movement up to 30 s after gentle stimulation, or untraceable larvae) and on behavioral disruption, i.e., convulsions correlated with the disability to bury themselves into the sediment. Moreover, we removed dead animals to avoid contamination of the test vessels by carcass decomposition products. Our criterion for test validity was a mortality rate of $\leq 10\%$ in the control.

## Chemical analyses

Thiacloprid analyses were conducted for water samples and larvae. The medium was analyzed to determine whether actual and nominal aqueous thiacloprid concentrations matched, and to confirm adsorption of thiacloprid to zeolites. For each tested concentration, one replicate was analyzed. Water samples were taken from the exposure experiments directly before larvae were introduced (0 h) and, additionally, at the end of the exposure (96 h). The samples were stored in 1.5 mL Eppendorf tubes (Eppendorf, Germany) at $-18$ °C. Prior to analysis, samples were centrifuged at 3,000 rpm for 3 min, and 1.0 mL supernatant was filtered through a PTFE syringe filter (pore size 0.45 $\mu$m, Chromafil; Macherey-Nagel, Düren, Germany). Thiacloprid concentrations were quantified by LC-MS. Internal thiacloprid concentrations of *C. riparius* larvae were determined for surviving individuals after termination of the exposure experiment. Animals originated from all

treatments, excluding those which were exposed to zeolites solely. Moreover, larvae were analyzed both with gut contents (these larvae were frozen directly in liquid nitrogen after the termination of the exposure) as well as with depleted guts. For gut depletion, larvae were transferred to one Petri dish per treatment which was filled with 150 mL filtered and dechlorinated tap water after termination of the exposure experiment. Petri dishes were partially wrapped in aluminum foil to reduce light-induced stress (*Hirabayashi & Wotton, 1998*). Under these conditions, the larvae were allowed to empty their guts for 24 h, subsequently frozen in liquid nitrogen, and stored at −80 °C. To quantify thiacloprid within the larvae, a liquid-liquid extraction method was developed based on the QuEChERS extraction procedure that originally was used in food analysis but recently adjusted to analyze environmental samples and biota (*Anastassiades et al., 2003*; *Berlioz-Barbier et al., 2014*). For sample extraction and cleanup, 10 –30 mg of frozen larvae tissue was homogenized in liquid nitrogen with a Micro-homogenizer (Carl Roth GmbH, Karlsruhe, Germany) in a 2.5 mL tubes (Eppendorf, Hamburg, Germany). A volume of 20 μL deuterated internal standard thiacloprid-D4 in methanol (Sigma-Aldrich, St. Louis, MO, USA) was added, resulting in a final concentration of 8.0 μg/L in the analyzed sample. Subsequently, 0.5 mL water (chromasolv; Sigma-Aldrich, St. Louis, MO, USA) and 0.5 mL acetonitrile were added, and the sample was shaken for 20 s using a Vortex mixer. After addition of 25 mg NaCl (Sigma-Aldrich, St. Louis, MO, USA) and 75 mg anhydrous $MgSO_4$ (Sigma-Aldrich, St. Louis, MO, USA) the sample was immediately shaken for 20 s using a Vortex mixer and centrifuged for 2 min at 10,000 rpm. A volume of 0.5 mL of the acetonitrile phase was transferred to 12 mg primary-secondary amine (PSA; Agilent Technologies, Waldbronn, Germany) and 90 mg anhydrous $MgSO_4$, then shaken for 20 s and centrifuged for 2 min at 10,000 rpm. The supernatant was concentrated in a nitrogen stream at room temperature, and the residue was reconstituted in 250 μL methanol and filtered with a PTFE syringe filter (pore size 0.45 μm, Chromafil; Macherey-Nagel, Düren, Germany). Also these samples were analyzed by LC-MS.

For LC-MS analysis, a 1260 Infinity LC system coupled to a 6550 iFunnel QTOF LC/MS system (Agilent Technologies, Waldbronn, Germany) was used. Aliquots of 10 μL were injected onto a Zorbax Eclipse Plus C18 column (2.1 × 150 mm, 3.5-Micron narrow bore; Agilent Technologies, Santa Clara, CA, USA) with a narrow bore guard column (2.1 × 12.5 mm, 5-Micron; Agilent Technologies, Santa Clara, CA, USA). A jetstream electrospray ionization (ESI) source was operated with a nebulizer pressure of 35 psig, drying gas temperature of 160 °C, a flow rate of 16 L/min, and a fragmentor voltage of 360 V. In the positive mode, capillary voltage was set to −4,000 V, skimmer voltage to 65 V, and the nozzle voltage to −500 V. The mass range was 80–1,200 m/z with a data acquisition rate of 1 spectrum/s. The sheath gas temperature was set to 325 °C with a flow rate of 11 L/min. For internal calibration, purine and HP0921 (m/z = 121.0508, 922.0097; Agilent Techologies, Santa Clara, CA, USA) were used, the mass range was set to 50 ppm. A gradient elution was performed at a flow rate of 0.3 mL/min using water and methanol both containing 0.1% formic acid. The initial content of 95% water was decreased after 1 min to 5% water over 7 min and, after another 7 min at 5%, increased to 95% water over 30 s. Data analysis was performed with MassHunter software (Agilent Technologies, Santa

Clara, CA, USA). For quantification of water-samples, a calibration curve in the range of 0.2-10 µg/L thiacloprid was used ($r^2 = 0.9991$), the limit of detection was 0.2 µg/L. Thiacloprid concentrations in larvae were calculated based on peak area of the deuterated internal standard of 8.0 µg/L and the limit of detection was 1.0 µg/L.

### LA-ICP-MS imaging

To verify ingestion of zeolites by the larvae, LA-ICP-MS imaging techniques for the detection of aluminum within thin sections of larvae were used. Therefore, 10 surviving larvae, exposed to the highest zeolite concentrations or the control, were fixed in 2% glutardialdehyde (Sigma Aldrich, Darmstadt, Germany) buffered in 0.005 M cacodylate (sodium cacodylate trihydrate, pH 7.4; Sigma Aldrich, Darmstadt, Germany). Fixed larvae were stored at 4 °C for at least one week. Afterwards, samples were de-calcified in 5% TCA (trichloroacetic acid, ≥99% p.a.; Carl Roth GmbH, Karlsruhe, Germany) in formol (37% p.a., stabilized with methanol; Carl Roth GmbH, Karlsruhe, Germany), dehydrated with ethanol and routinely processed for Technovit® embedding (Technovit® 7100; Heraeus Kulzer GmbH, Germany). Embedded samples were cut by a rotation microtome (Leica RM2265; Leica Biosystems Nussloch GmbH, Wetzlar, Germany) into sections of 7 µm thickness which were used for LA-ICP-MS imaging experiments. These were performed with the LA system LSX 213 G2$^+$ (CETAC Technologies, Omaha, NE, USA) incorporating a frequency quintupled Nd:YAG laser with a wavelength of 213 nm and equipped with Chromium 2.2 software. The laser system device was coupled to a triple quadrupole based inductively coupled plasma mass spectrometer (iCap TQ; Thermo Fisher Scientific, Darmstadt, Germany). The LA parameters relating to spot size, scanning speed, laser energy and carrier gas flow were optimized based on the best signal-to-noise ratio in combination with highest spatial resolution. The samples were ablated using a line by line scan with a laser energy density of 7,55 J/cm$^2$, 20 Hz laser shot frequency, 50 µm spot size and 250 µm/ s scan speed. The aerosol was transported to the ICP with a carrier gas of helium (0.45 L/min) passing the ablation chamber. For maximum sensitivity and to minimize possible interferences, the measurement was performed in kinetic energy discrimination mode (KED) with helium as cell gas. $^{27}$Al was monitored with dwell times of 0.1 s. Data evaluation was performed using the software ImageJ (National Institutes of Health, Bethesda, MD, USA).

### Data analyses

A generalized linear mixed model (GLMM) with binary data distribution was used, with the Petri dish identity as a random factor. The statistics were done with "R" (*R Development Core Team, 2017*) and an α- level set to 0.05. SigmaPlot 13 (Systat Software) was used to plot graphs.

## RESULTS

### Zeolite characteristics

The results of the particle size measurements by laser diffraction in the aqueous phase are shown in Table 1. As expected, the sizes of Y30 particles decreased with the processing

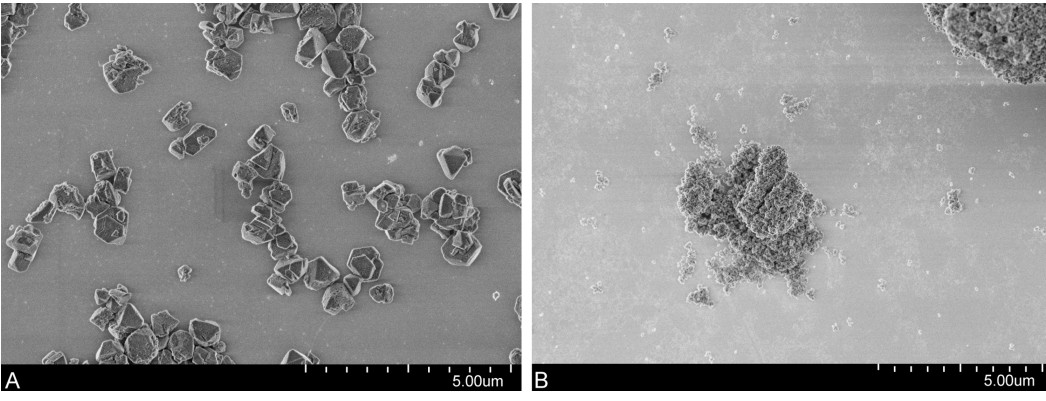

**Figure 1** **SEM images.** (A) Y30 zeolites (after milling and settling) and (B) calcinated H-Beta(OH)-III zeolites.

**Table 1** **Particle size distribution of differently processed Y30 zeolites and Beta(OH)–III zeolites, obtained by laser diffraction measurements.** $d_{10}$, $d_{50}$ and $d_{90}$ are particle size distribution parameters, indicating that 10, 50 or 90% [v/v] of all particles were smaller or equal to the indicated size.

|  | $d_{10}$ [nm] | $d_{50}$ [nm] | $d_{90}$ [nm] |
|---|---|---|---|
| Commercial Y30 | 1,082 | 2,772 | 5,838 |
| Y30 milled | 603 | 1,552 | 4,401 |
| Y30 milled, supernatant after settling[a] | 202 | 441 | 691 |
| Beta(OH)–III (non-calcinated) | 3,700 | 21,500 | 108,500 |
| H-Beta(OH)–III (calcinated)[a] | 4,100 | 9,200 | 20,900 |

**Notes.**
[a] The particles which were used in the acute toxicity tests

of the particles and matched well with the SEM results (Fig. 1A). The measured particle size of Beta(OH)-III zeolites, however, was several micrometers. Although calcination reduced the particle size or aggregation/agglomeration, the particles were much larger than 50 nm, the reported particle size of Beta(OH)-III zeolite primary particles. The comparison of Y30 and H-Beta(OH)-III zeolites in SEM measurements revealed that Y30 zeolites, processed by milling and sedimentation, formed single primary particles (Fig. 1A). In contrast, H-Beta(OH)-III formed much larger aggregates/agglomerates of primary particles (Fig. 1B). The sorption experiments showed that sorption of thiacloprid on Y30 was strong and followed the Langmuir isotherm model. The $K_d$ value at low concentrations within the linear range of the sorption isotherm was 82,500 L/kg.

## Chemical analyses

The chemical analyses of the medium revealed the thiacloprid concentrations did not differ between the start and the end of the experiment. Moreover, it was shown that both zeolite types strongly adsorbed thiacloprid. Y30 particles in the toxicity assays adsorbed thiacloprid in the predicted manner, hence, the concentration of thiacloprid in mixtures was below the detection limit (<0.2 µg/L) when high Y30 concentrations were added (97% adsorption was calculated), at 0.5 µg/L when the medium Y30 concentration was added (60% adsorption was calculated) and at 0.65 µg/L for the lowest Y30 concentration

(30% adsorption was calculated). H-Beta(OH)-III particles, however, adsorbed thiacloprid to a much higher extent, so that dissolved thiacloprid was not detectable in all mixture experiments containing H-Beta(OH)-III zeolites. When applied solely, the thiacloprid concentration was $1.0 \pm 0.09$ µg/L. The concentrations of thiacloprid associated with larval tissue were similar for all larvae exposed to thiacloprid, solely or in mixtures (58.41 ng/g $\pm$ 14.42 ng/g (wet weight) on the average), irrespective of the zeolite type added or the presence of gut contents in the sample.

## Acute toxicity test

In general, control animals did not show any anomalies in behavior, and the average mortality rate was $4 \pm 8\%$ (Figs. 2 and 3). Consequently, the criterion of validity (mean mortality $\leq 10\%$ in the control) was met. Moreover, both zeolites, applied as single substances, did not lead to behavioral disorders. In contrast, larvae exposed to thiacloprid as single compound or to mixtures with low or medium Y30 concentrations displayed behavioral impairments (Fig. 2). However, behavioral changes were found to be delayed in a concentration-dependent manner of Y30. Moreover, larvae exposed to the treatment with the highest Y30 concentration and, thus, highest sorption of thiacloprid did not show any behavioral disruption. In the experiment with H-Beta(OH)-III particles, no animals with behavioral disorders were observed. Experiments with thiacloprid solely showed an almost identical increase of convulsions over time as larvae of the identical treatment type in the experiment with Y30 did. Mortality predominantly occurred between 72 and 96 h of exposure. In the experiment with Y30, we did not find any statistical differences between animals exposed to Y30 solely and the control (GLMM, $df = 3$, $F = 0.5351$; $p$-values for comparisons between the control and low Y30 concentration: 0.676, medium concentration: 0.467, high concentration: 0.676). Nevertheless, larvae exposed to thiacloprid, solely or in the mixtures that comprised low and medium Y30 concentrations, suffered from significantly increased mortality rates compared to the control or the mixture with the highest Y30 concentration, but were not significantly different from one another (GLMM, $df = 4$, $F = 10.849$; $p$-values for the comparison between the control and thiacloprid solely, mix low and mix medium: <0.001, mix high: 0.4162; $p$-values for the comparisons between thiacloprid solely and mix low: 0.4314, mix medium: 0.0819, mix high: <0.001; $p$-values for the comparison between mix high and mix low, mix medium and thiacloprid solely: <0.001). However, the mean mortality rate decreased with increasing zeolite concentration (Fig. 3A). Larvae exposed to the mixture with the highest Y30 concentration showed a low mortality rate that did not differ to the control. Likewise, no statistical differences between animals solely exposed to H-Beta(OH)-III zeolites and the control were found (GLMM, $df = 3$, $F = 0.3354$; $p$ values for comparisons between the control and low H-Beta-OH-III concentration: 1.0; medium concentration: 0.651, high concentration: 0.414). In further analyses, it was shown that the mortality rate of larvae exposed to the mixtures of thiacloprid and H-Beta(OH)-III did not differ from the control as well. In the contrary, animals exposed solely to 1.0 µg/L thiacloprid exhibited significantly higher mortality rates than larvae exposed to the mixtures or the control (GLMM, $df = 4$, $F = 13.488$; $p$-value for the comparison between the control

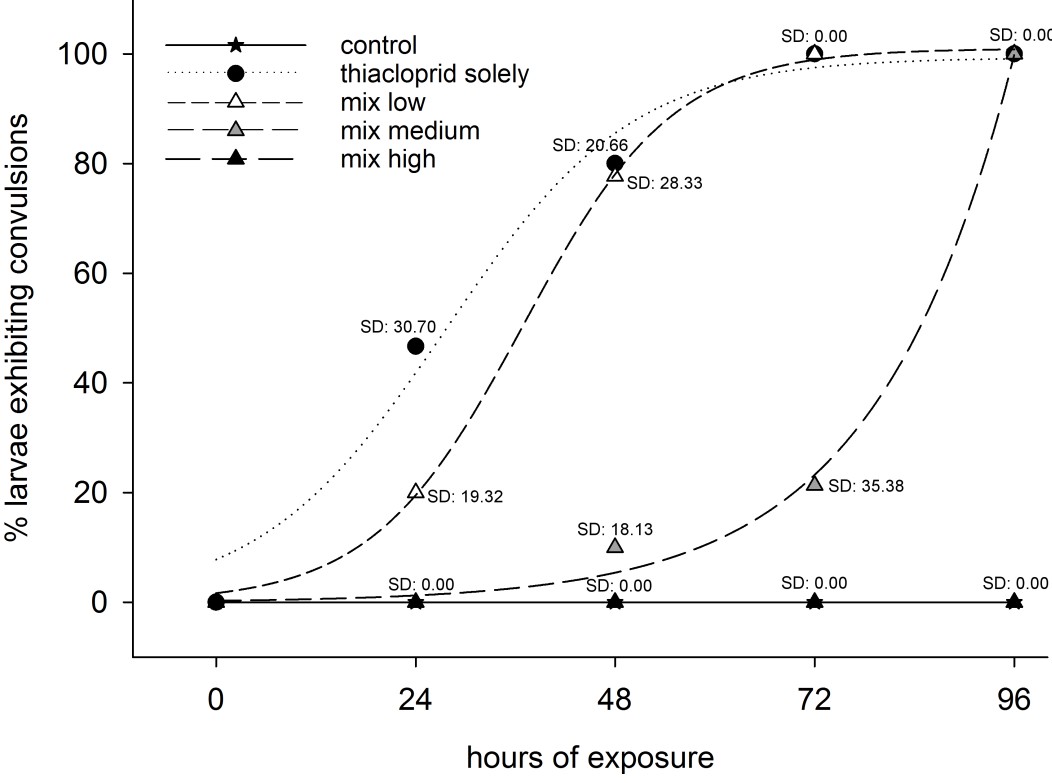

**Figure 2  Behavioral disorders.** Best fitting curves, representing the percentages of living *C. riparius* larvae exhibiting convulsions in the experiment with Y30 zeolite. Animals were exposed to dechlorinated and filtered tap water as control, to 1.0 µg/L thiacloprid solely, or in mixtures with low, medium or high Y30 concentrations (per test vessel, $n = 15$).

and thiacloprid solely: <0.001, mix low: 0.433, mix medium: 0.660 and high: 0.660; *p*-values for all comparisons between thiacloprid solely and all three mixtures was <0.001; Fig. 3B).

## LA-ICP-MS imaging

To evaluate whether zeolites were taken up by the larvae, LA-ICP-MS imaging technology with aluminum as marker for zeolites was used. The distribution of aluminum within the larvae confirmed that both zeolites were ingested by the larvae and were widely present in the gut lumen (Fig. 4). However, there was no indication that zeolites were taken up by the tissues of the organism, not even by the resorptive gut epithelium. As expected, larvae of the control did not show any aluminum-containing particles in the gut lumen or in any tissue.

## DISCUSSION

The purpose of this study was to assess the acute toxicity of two nano-zeolites and to test whether adsorption of the insecticide thiacloprid to these zeolites alters its acute toxicity. Since the main scope was on the possible interactions between thiacloprid and zeolites and not particularly on the assessment of distinct field situations, we did not use environmentally relevant concentrations, particularly for the nanoparticles. Our results

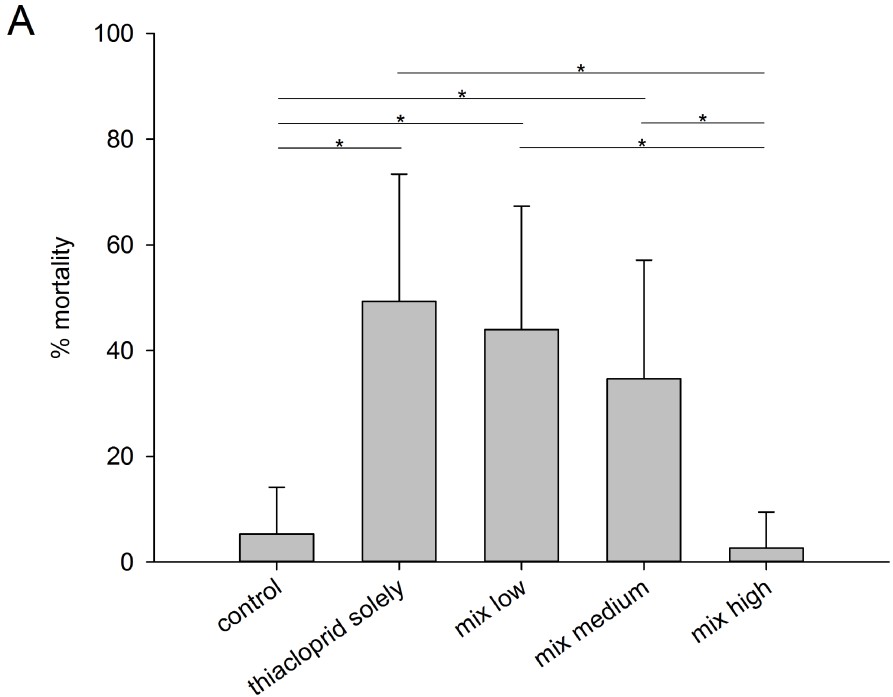

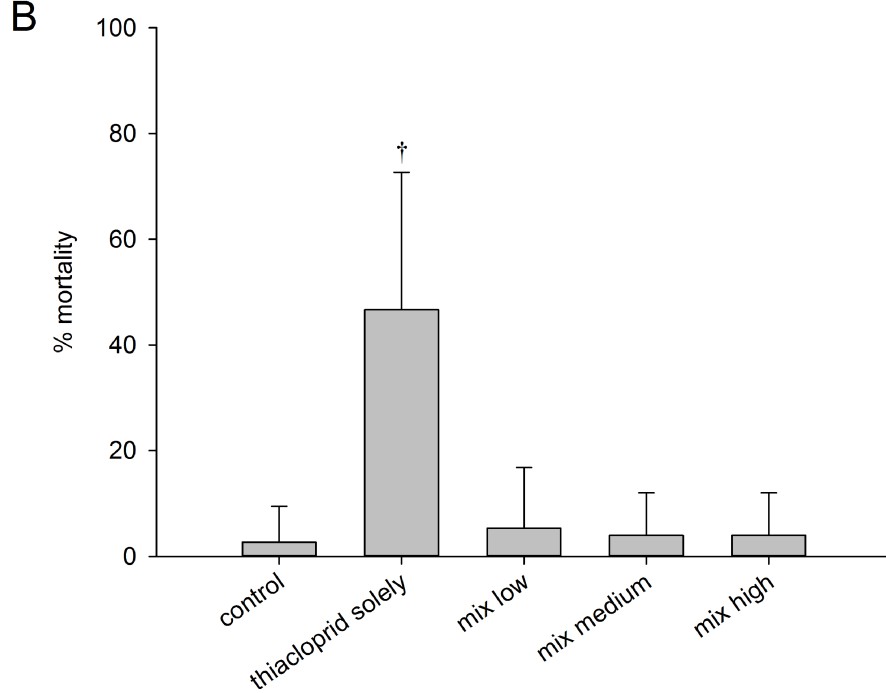

**Figure 3  Mortality rates.** Means mortality rates ±SD of *C. riparius* larvae per test vessel ($n = 15$) after exposure for 96 h to control conditions, to 1.0 µg/L thiacloprid solely, or to 1.0 µg/L thiacloprid in addition of low, medium and high concentrations of zeolites. (A) Experiment with Y30 zeolites. Asterisks (*) mark $p \leq \alpha$. (B) Experiment with H-Beta(OH)-III zeolites. The cross (†) marks the treatment group that showed significant differences vs. all other treatments.

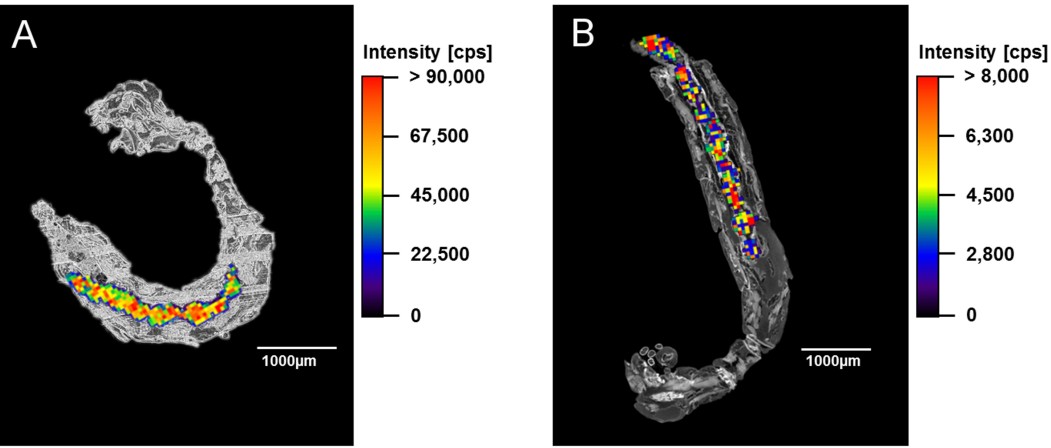

**Figure 4** **LA-ICP-MS analyses.** Overlay of images of a 7 μm thick section of a *C. riparius* larva exposed to the highest zeolite concentration solely and LA-ICP-MS images, which show the distribution of aluminum within the larva. (A) Larva exposed to Y30 zeolites. (B) Larva exposed to H-Beta(OH)-III zeolites.

showed that both nano-zeolites, even in the rather high test concentration, did not affect the chosen endpoints when applied solely. This is in accordance with published toxicity studies on larger zeolites, for which save applications (e.g., in water remediation) have been shown (*Lehman & Larsen, 2014*). Unfortunately, and contrary to our expectations, H-Beta(OH)-III particles could not be tested in their nano-sized form, since the small primary particles formed rather large aggregates/agglomerates in aqueous dispersions. Such formation of aggregations or agglomerations is a common issue in various studies dealing with colloids/nanoparticles and thus an additional challenge in exposure assessment (*Savolainen et al., 2010*; *Schultz et al., 2015*). In our case, we additionally expect that the H-Beta(OH)-III aggregates/agglomerates increased in size with time of exposure and thus assume that larvae were gradually exposed to smaller aggregates/agglomerates during the beginning of the experiment (due to previous ultrasonication) and to larger aggregates/agglomerates at the end of the exposure phase. Consequently, larvae have to be considered to be exposed to varying sizes of H-Beta(OH)-III particles with time, but some of these particles were neither nanoparticles, nor colloids. Consequently, since the size of primary particles is less important than the sizes of the aggregations (*Seipenbusch et al., 2010*), it is highly likely that the large sizes of the aggregates/agglomerates impaired possible cell uptake or, at least, largely reduced its probability. In fact, no cellular uptake of zeolite particles was detected by LA-ICP-MS, neither for nano-sized Y30 nor for the larger H-Beta(OH)-III particles, even though zeolites were definitely ingested by the larvae. The ingestion of zeolites by larvae is very plausible, since the zeolites deposit on the substratum—larger particle aggregates faster than smaller single crystal particles (e.g., *Schultz et al., 2015*; *Scown, Van Aerle & Tyler, 2010*)—and thus can be incorporated by the sediment-dwelling and detritus-feeding larvae of *C. riparius* (*Armitage, Cranston & Pinder, 1995*). Although we found no indication that these ingested particles exerted any toxic effect the results might be different under chronic exposure conditions. For example, *Zhu, Chang & Chen (2010)* showed that TiO$_2$ nanoparticles accumulated in the intestinal tract

of *Daphnia magna* and these animals were unable to empty their guts in the usual time. Therefore, *Zhu, Chang & Chen (2010)* hypothesized negative impacts of this nanoparticle accumulation, e.g., on growth rate or reproduction, under long-time exposure. So, we cannot exclude such effect for chronically exposed chironomid larvae as well. In contrast to the tested zeolites, thiacloprid, applied as a single substance, affected both investigated biological endpoints significantly in both experiments. Specifically, thiacloprid led to heavy convulsions of larvae (associated with the inability of burying themselves into the sediment) and an increased mortality rate. Both effects, convulsions and mortality of larvae, can be assigned to the mode of action of thiacloprid, which stimulates the insect's nervous system by acting as agonist of the nicotinic acetylcholine receptor (*Elbert et al., 2008*), and have been already described in former studies (e.g., *Langer-Jaesrich, Köhler & Gerhardt, 2010*). Furthermore, our results corroborate the findings of a previous study by *Lorenz et al. (2017)*, where, among others, fourth instar larvae of *C. riparius* where exposed to 1.0 µg/L thiacloprid in an identical experimental setup. However, although the same thiacloprid concentration was used in the mixture experiments, we found that Y30 zeolites reduced the acute toxic effects of thiacloprid in a concentration-dependent manner, and that all animals exposed to mixtures including H-Beta(OH)-III did not show any increased mortality or behavioral impairment compared to the control. Our results suggest that the investigated zeolites do not act as a vehicle for transporting thiacloprid into the cells of *C. riparius* larvae, and that the different size of the particles does not have any effect in the tested size range. We furthermore conclude that the insecticide's toxicity decreased due to the lower bioavailability of thiacloprid caused by adsorption, as the results of the acute toxicity tests correlate well with the measured equilibrium concentrations of aqueous dissolved thiacloprid. A reduction in the toxicity of chemicals caused by adsorption on particles is well documented in the literature (e.g., *Knauer, Sobek & Bucheli, 2007*; *Koelmans et al., 2006*). For example, *Baun et al. (2008)* have demonstrated that nano-$C_{60}$ particles sorb pentachlorophenol (PCP) significantly, which leads to reduced toxicity of PCP on *Daphnia magna* when the animals were exposed to both substances simultaneously. Furthermore, several studies described adsorption of chemicals on zeolites (e.g., *Braschi et al., 2010*; *Ellis & Korth, 1993*; *Ötker & Akmehmet-Balcioğlu, 2005*) and, for instance, *James & Sampath (1999)* have reported a decrease cadmium concentrations in water and fish (*Oreochromis mossambicus*) in the presence of zeolites. However, the mortality rate of larvae exposed to a mixture of thiacloprid and medium Y30 concentrations was higher than expected on the basis of a previous study (*Lorenz et al., 2017*). There, larvae had been exposed, amongst other concentrations, to 0.4 µg/L thiacloprid in an identical experimental setup, which resulted in a mortality rate of approximately 13%. Furthermore, no behavioral impairments were observed. In the present study, medium concentrations of Y30 lead to a similar aqueous thiacloprid concentration, but the mortality rate was roughly 2.5-times higher and all larvae showed abnormal behavior after 96 h of exposure. This disparity indicates that thiacloprid might desorb from the nanoparticles to some minor extent under the different physicochemical conditions in the lumen of the intestinal tract. However, the amount of desorbed thiacloprid must have been too low to cause negative effects when a greater portion of thiacloprid was adsorbed in the experiment with high Y30

concentrations. Our conclusion that thiacloprid desorbed from zeolites in the gut lumen is also supported by the chemical analyses of the larvae's internal thiacloprid concentrations. Here, similar internal thiacloprid concentrations were found in all animals - irrespective of the zeolite type or concentration they were exposed to and irrespective of whether they had been allowed to defecate or not. In this context, we feel that further research is needed to explain this uniformity in the internal pesticide concentrations, particularly since it has been shown that the toxicity of thiacloprid on *C. riparius* can also be reduced/delayed by non-adsorbing particles (*Lorenz et al., 2017*).

In our study, we have documented the mechanistic basis for the reduced acute toxicity of thiacloprid when interacting with zeolites. In the future, chronic effects of both nano-zeolites used in this study should be tested to draw conclusions about their ecotoxicity. Further studies should also investigate the protective effects of zeolites in environmentally relevant concentrations, even though the quantification of these concentrations still remains a challenging issue. However, due to the high adsorption capacity that zeolites can have, measurable effects of realistic zeolite concentrations present in the environment might be likely.

## ACKNOWLEDGEMENTS

We thank Leilei Luo, Reiner Anwander, Yucang Liang, Martin Pattky, Christian Zwiener, Sylvain Merel, Boris Bugsel, Klaus Roehler, Sandra Dietz, Elisabeth Früh, Sang Hyun Ahn, and Suk Bong Hong for discussions, helping advice, and support. We also thank Miriam Langer for reviewing this manuscript and for her constructive comments.

### Funding
The study was part of the intramural graduate program "EXPAND" funded by the Excellence Initiative of Eberhard Karls University Tübingen, Germany. Beta(OH) III nanoparticles were synthesized and provided by the National Creative Research Initiative Program (2012R1A3A2048833) through the National Research Foundation of Korea funded by the Korean government (MSIP). Publishing was supported by the Deutsche Forschungsgemeinschaft and the Open Access Publishing Fund of University of Tübingen. The funders had no role in study design, data collection and analysis, decision to publish, or preparation of the manuscript.

### Grant Disclosures
The following grant information was disclosed by the authors:
Excellence Initiative of Eberhard Karls University Tübingen, Germany.
Korean government (MSIP).
Deutsche Forschungsgemeinschaft and the Open Access Publishing Fund of University of Tübingen.

## Competing Interests

The authors declare there are no competing interests.

## Author Contributions

- Carla S. Lorenz conceived and designed the experiments, performed the experiments, analyzed the data, wrote the paper, prepared figures and/or tables.
- Anna-Jorina Wicht, Leyla Guluzada and Barbara Crone performed the experiments.
- Uwe Karst, Rita Triebskorn, Stefan B. Haderlein and Carolin Huhn reviewed drafts of the paper.
- Hwa Jun Lee contributed reagents/materials/analysis tools, reviewed drafts of the paper.
- Heinz-R. Köhler conceived and designed the experiments, reviewed drafts of the paper.

## Supplemental Information

Supplemental information for this article can be found online at http://dx.doi.org/10.7717/peerj.3525#supplemental-information.

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
