# Peer review of "Nano-sized zeolites as modulators of thiacloprid toxicity on Chironomus riparius"

_PeerJ, doi:10.7717/peerj.3525_

## Round 0.1 · original submission · Minor Revisions

Hi,

The first reviewer has invested substantial effort to go through the manuscript. Please consider his/her comments carefully. Given that a lot of them are quite specific, it should be straightforward to take them on board, should you decide to do so. Otherwise, please let us know why you decided to not include them.

Best wishes
Thomas

·

Basic reporting

The authors have investigated the acute effects of 2 different zeolites Y30 and Beta(OH)-III zeolite (stated as Nano-sized in the heading and abstract) solely in different concentrations on the sediment dwelling organism Chironomus riparius using the endpoints mortality and behaviour (convulsions and accompanying by decreased burring behaviour). They also investigated the acute effects of the neonicotinoid Thiacloprid solely and in combination with different zeolite concentrations. It was their aim to understand if the zeolites are acute toxic to C. riparius and if the zeolites would modulate the acute toxicity of Thiacloprid. They were interested if the zeolites would increase the uptake or if they would reduce the bioavailability of Thiacloprid both altering the toxicity of Thiacloprid. They are asking additionally if the particle size would influence of the sorption capacity of the zeolites.
Methodically they described the used zeolites Y30 and Beta(OH)-III and their treatment as well as the sizes measurements. They investigated the sorption characteristics of one of the Zeolites (Y30) towards Thiacloprid and derived different sorption percentages and thus different zeolite concentrations called latter low, medium and high. For Beta(OH)-III they used the concentrations from the Y30 zeolite and did not determine a specific sorption capacity. They describe following points in the material section:
• the solution preparation
• Test organisms
• performance of the acute toxicity test
• chemical Thiacloprid analysis in water and in the larvae (including gut content or without gut content)
• use of LA-ICP-MS to determine the ingestion and uptake of zeolites in the C. riparius larvae.
• Statistics
In the result part the authors show that Y30 behaves as nanoparticle, but that Beta(OH)-III formed larger aggregates which are above the nanoparticle scale. In the chemical analysis they found constant concentration of solely Thiacloprid, but they also demonstrate that both zeolites strongly absorb Thiacloprid. The find that absorption of Thiacloprid at Y30 is following a predicted manner having in the low Y30 concentration a higher Thiacloprid concentration than in the medium Y30 concentrations and so on. For Beta(OH)-III they report a much higher sorption potential resulting in Thiacloprid always below LOD independently of the Beta(OH)-III concentration. Interestingly the authors find in larval tissue the same Thiacloprid concentration independently of the experimental design.
They determine normal control mortality and normal behaviour, and the tests should be seen as valid. They do not find an acute effect of both applied single zeolites neither on mortality nor on behaviour (which is not surprising since the zeolites would not exert the same mode of action than the neonicotinoid Thiacloprid). Only in the solely Thiacloprid treatment or Thiacloprid mix with low and medium Y30 concentrations increased mortality was detected. Behaviour changes was detected in those treatments in a time dependent way. For Beta(OH)-III in combination with Thiacloprid neither mortality nor behaviour changes were detected. In the LA-ICP-MS analysis the authors found that both zeolites were ingested but found no evidence that there occurred uptake to other tissues.
In the discussion part the authors state that the used concentrations of zeolites might not resemble environmental relevant concentrations, but that even at high concentrations no negative acute effects could have been detected. But they also point out that for H Beta(OH)-III they are above the nano-size scale, since larger aggregates have been formed (which is a common challenge in nano particle studies), which probably varied in size over the exposure time.
The authors are aware that the findings of their acute study cannot be transferred to a chronic setup. Derived from their results the authors state, that the investigated zeolites did not act as vehicles for Thiacloprid in the cells and that the different tested sizes of particles have not an effect on toxicity. Contrary they find that the toxicity is decreased by lower bioavailability of Thiacloprid. They point out critically that in a previous study by Lorenz et al. 2017 with a similar set up the mortality in the actual study was by 2.5 times higher (Unfortunately the study is only accepted and not available at the moment for comparison). They speculate that the difference in the studies might be due to desorption potential in the gut of C. riparius. They suggest to investigate in further studies why there was no differences of internal Thiacloprid concentration in C. riparius in the different tested treatments.

Overall the study is clear and unambiguous written. The English language is well applied and easy to understand. References and background of the topics are sufficiently provided. The structure of the article is appropriate and figures and tables are supplied and easy to read. Although the table and figure captures were missing and should be provided in the future manuscript. Research questions are stated clearly and answered with the results. The raw data are available and fully labelled, although some typos should be corrected. The raw data result in the same picture like the text as well as the figures and tables. Although I am critical about the statistical approach using all 75 organisms per treatment. To my knowledge the replicates (n=15 a 5 organisms ) per treatments should be used for their statistical analysis. If this is just a misunderstanding the authors should specify their approach.

Experimental design

The findings of the study is relevant for the field of Nano-particle investigations and is within the aims and scopes of PeerJ. The asked questions as well as the answers derived from the study are significant for the risk assessment of Nano-particles. The authors recognise the boundaries of their study and communicates them (results from acute study cannot extrapolated to a chronic study). They supply different investigations beside the acute test like the chemical analysis as well as zeolites fate in the in the organism. Various techniques are applied to answer this topics (Thiacloprid analysis in water and in the organisms by LC-MS, LA-ICP-Ms imaging, Size measurements and sorption experiments of the zeolites.) The methods are, as far as I have experience in the field, well described and should be sufficient to repeat the experiment. Tthe approach for statistical analysis should be revised or clear communicated.

Validity of the findings

The authors do communicate the boundaries of their study (acute and chronic, Beta(OH)-III zeolite agglomeration). They also point out which details has to be addressed in future studies (same Thiacloprid concentrations in tissue independently of treatment). Negative results are transparently reported and explained. negative findings do not oppose the positive findings.
Data are plausible and the control values (positive or negative, as well as repeated treatments) complete the general impression. Number of replicates are sufficient make their statement for the tested zeolites. As mentioned above their statistic should be changed based on a replicate approach. If the statistical approach was not well communicated this should be improved.
Conclusions are well stated and answer the research questions asked in the introduction. Only direct findings are included in the conclusions.

Additional comments

• In the heading as well as in the abstract the authors state that Nano-sized zeolites are investigated, although one of the investigated zeolites Beta(OH)-III was larger and even formed aggregates (p 14, line 326). This findings should be mentioned in the abstract, since it is an important part.
• In the materials section it should be clearly stated which type of the processed zeolites are used in the acute toxicity test and for the SEM (Y30 milled, supernatant after settling and H-Beta(OH) III (calcinated)). In the present manuscript the word final is used, which has not been defined.
• Page 7 Line 151: Please introduce what the Kd value is, when it is first introduced in the manuscript.
• Page 7, Line 151: Why have the authors only investigated the sorption characteristics and Kd value of the Y30 zeolite? As their results also show the sorption characteristics of the H-Beta(OH) III zeolite behaves differently with a much higher sorption rate.
• P15, line 211. The authors state that low absorption of Thiacloprid was expected. Is anything known about the behaviour of zeolites in glass. Have the zeolites solutions also used for saturation?
• P15, line 212: 30 g sand , dry or wet weight?
• P15, line 212: The authors state that adsorption was negligible of the tested substances. Is this known for the tested zeolites? Please specify this point, preferably with a reference. To the reviewers small knowledge about Nano-particles, it is quite possible that Nano particles may bind to sand.
• After ultra-sonication of the test solutions to have an equally distributed solution, the test run 96h, there a settling of the zeolites is likely.
• P13, line 314: as stated above it is critical to use the statistical approach including all 75 organisms per treatment. To my knowledge the 15 replicates (n=15 a 5 organisms ) per treatments should be used for the statistical analysis. If the statistical approach was not well communicated this should be improved.
• Page 14, line 333: The constant concentration of Thiacloprid could be stated also in the raw data: But this is not a critical issue. Also the Thiacloprid concentration in the larval internal tissue could be stated there.
• Page 17, line 412: Please acute before toxic to avoid misunderstanding.
• Table and figure caption are missing. This should be absolutely provided in the revised article.
• Table 1: For a better understanding the authors could highlight the final used particles in the test: In the table caption please declare what is meant by d10, d50 and d90
• In Fig 2 the variation (e.g. Standard deviation) should be included. Otherwise a limited perspective might arise.
• Fig 3. Alternatively a different graph type like box plots could be considered for better data overview
• Raw data: sheet for Beta(OH)-III zeolite mortality: here the label in column A is not correct. Please replace Y30 by Beta(OH)-III. In sheet for Beta(OH)-III zeolite behaviour at 96h (column x and Y) – please adapt values to % numbers like in previous columns.

Reviewer 2 ·

Basic reporting

No comment. Very good manuscript.

Experimental design

No comment.

Validity of the findings

No comment.

Additional comments

Well designed study, opening up the current nanoparticle research to more complex setting such as mixture toxicity which is needed to fully understand and predict nanomaterial hazards.

---

## Round 0.2 · accepted · Accept

Dear all,

Thanks for the detailed reactions to the reviewer's comments. I would suggest to move forward with the publication as is.

Best wishes
Thomas